# Infections as Novel Risk Factors of Atherosclerotic Cardiovascular Diseases: Pathophysiological Links and Therapeutic Implications

**DOI:** 10.3390/jcm10122539

**Published:** 2021-06-08

**Authors:** Piotr Szwed, Aleksandra Gąsecka, Mateusz Zawadka, Ceren Eyileten, Marek Postuła, Tomasz Mazurek, Łukasz Szarpak, Krzysztof J. Filipiak

**Affiliations:** 11st Chair and Department of Cardiology, Medical University of Warsaw, 02-097 Warsaw, Poland; szwedp12@gmail.com (P.S.); tmazurek@kardia.edu.pl (T.M.); krzysztof.filipiak@wum.edu.pl (K.J.F.); 22nd Department of Anaestesiology and Intensive Therapy, Medical University of Warsaw, 02-097 Warsaw, Poland; m.t.zawadka@gmail.com; 3Department of Experimental and Clinical Pharmacology, Center for Preclinical Research and Technology Medical University of Warsaw, 02-097 Warsaw, Poland; ceren.eyileten-postula@wum.edu.pl (C.E.); mpostula@wum.edu.pl (M.P.); 4Institute of Outcomes Research, Maria Sklodowska-Curie Medical Academy in Warsaw, 03-411 Warsaw, Poland; lukasz.szarpak@gmail.com; 5Maria Sklodowska-Curie Bialystok Oncology Center, 02-034 Bialystok, Poland

**Keywords:** atherosclerosis, cardiovascular disease, infections, outcomes, risk factors

## Abstract

Atherosclerotic cardiovascular diseases (ASCVD) are the major cause of mortality worldwide. Despite the continuous progress in ASCVD therapy, the residual risk persists beyond the management of traditional risk factors. Several infections including *Helicobacter pylori* infection, periodontal disease, and viral infections are associated with the increased risk of ASCVD, both directly by damage to the heart muscle and vasculature, and indirectly by triggering a systemic proinflammatory state. Hence, beyond the optimal management of the traditional ASCVD risk factors, infections should be considered as an important non-classical risk factor to enable early diagnosis and appropriate treatment. Here, we summarized the currently available evidence regarding the role of inflammation in ASCVD and the association between the particular infections and pathogens (*Helicobacter pylori*, periodontal disease, pneumonia, *Cytomegalovirus*, *Human immunodeficiency virus*, *Herpes simplex virus*, and *severe acute respiratory syndrome coronavirus 2*) on the development and progression of ASCVD. We also speculated about the potential therapeutic implications of the anti-inflammatory and anti-infective drugs on ASCVD outcomes, including drugs routinely administered in patients with ASCVD (statins, P2Y12 receptor inhibitors, and angiotensin-converting enzyme inhibitors/angiotensin receptor blockers) and novel strategies aiming at residual risk reduction (colchicine, anti-cytokine drugs, and methotrexate). Considering the emerging association between infections and ASCVD, it is crucial to determine the possible advantages of infection prevention and treatment in patients with ASCVD.

## 1. Introduction

Cardiovascular diseases (CVD) are the leading cause of mortality worldwide, accounting for an estimated 17.9 million cases of death per year [1]. Among them, the largest group is atherosclerotic CVD (ASCVD). Despite the continuous progress in ASCVD therapy, the residual risk persists beyond the management of traditional risk factors. Hence, there is still a need for a better comprehension of pathophysiological processes underlying ASCVD [2,3]. In addition to traditional risk factors such as smoking, visceral adiposity, diabetes, hypertension, and dyslipidemia, chronic inflammation is one of the well-established non-classical risk factors of atherosclerosis [4,5]. Prolonged exposure to oxidative stress leads to the activation of multiple cell types involved in inflammation, including leukocytes, platelets, endothelial cells, and smooth muscle cells. Cellular mediators released from these cells induce and sustain the chronic inflammation of the vessel wall, which is a crucial pathophysiological mechanism of atherosclerosis [6]. Vessel wall inflammation is also due to impaired production or activity of nitric oxide (NO), which leads to endothelial dysfunction at an early stage of atherosclerosis. Impaired NO release along with increased NO degradation leads to an imbalance between vasoconstriction and vasodilation and initiates numerous mechanisms that promote and exacerbate atherosclerosis [7].

Since infections are inevitably associated with activation of immune response and inflammatory processes, the relation between ASCVD and infections has been extensively studied. There is evidence that particular infections are a risk factor for the development and/or aggravation of ASCVD. In particular, the association of ASCVD with periodontal disease, *Helicobacter pylori (H. pylori)*, *Cytomegalovirus* (*CMV*), pneumonia, *Human immunodeficiency virus* (*HIV*), *Herpes simplex virus (HSV)*, and most recently *severe acute respiratory syndrome coronavirus-2* (*SARS-CoV-2*) infection has been described [8,9,10,11,12,13,14]. However, the pathomechanism of these associations is not entirely clear and hitherto research has provided controversial results. Considering that ASCVD are the major cause of death in high-income countries, whereas infections (a potentially modifiable risk factor) are the major cause of death in middle- and low-income countries [15], it is crucial to determine both the pathophysiological associations and the possible advantages of infection treatment in the course of ASCVD. This may have a major effect not only on global human health but also on the healthcare costs related to infections and ASCVD. Here, we summarized the currently available evidence on the role of infections in ASCVD development, considering (i) the role of inflammation in ASCVD, (ii) the impact of particular infections and pathogens (*Helicobacter pylori*, periodontal disease, pneumonia, *CMV, HCV, HIV*, *HSV*, and most recently *SARS-CoV-2)* on the development and progression of ASCVD and (iii) possible therapeutic implications, focusing on the anti-inflammatory and anti-infective efficacy of statins, P2Y12 inhibitors, angiotensin-converting enzyme inhibitors (ACE-I) and angiotensin receptor blockers (ARBs), colchicine, anti-cytokine drugs, and methotrexate.

## 2. The Role of Inflammation in Cardiovascular Diseases

Atherosclerosis is an inflammatory disease that underlies ASCVD and related complications, including acute coronary syndrome, ischemic stroke, and peripheral artery disease [6]. Cells involved in the development of atherosclerosis include macrophages/foam cells, endothelial cells, smooth muscle cells (SMC), T-lymphocytes, and platelets. In addition, extracellular vesicles (EVs) and microRNAs (miRNAs) released from these cells induce and sustain atherosclerotic processes in the vessel wall. The pathophysiology of atherosclerotic plaque formation is shown in Figure 1.

Accumulation of circulating low-density lipoproteins (LDL) in the subendothelial layer of the arteries is the initial stage in the development of atherosclerosis. Following endothelial injury, LDL adhere to extracellular matrix proteins exposed by the activated endothelial cells [16]. As LDL is trapped in the vascular wall, it undergoes oxidation by locally secreted reactive oxygen species (ROS), being transformed into oxidized LDL (ox-LDL) [17]. Ox-LDL, along with native LDL, are taken up by macrophages via scavenger receptors (SR), leading to macrophage transformation into foam cells [18]. Both macrophages and foam cells release a variety of proinflammatory cytokines such as soluble CD40 ligand, interleukin-1 (IL-1), IL-3, IL-6, IL-8, IL-18, and tumor necrosis factor-alpha (TNF-α) [19]. IL-1β, IL-6, and TNF-α stimulate the liver production of C-reactive protein (CRP), the most common inflammatory biomarker [20].

Besides macrophages/foam cells, endothelial cells are also involved in atherosclerotic plaque formation. After exposure to damaging factors, such as dyslipidemia, smoking, hypertension, or viruses, endothelial cells upregulate the transcription of nuclear factor-κB (NF-κB) and expose molecules that enhance leukocyte adhesion to the endothelium, such as P-selectin, E-selectin, vascular cell adhesion molecule-1 (VCAM-1), and intercellular adhesion molecule-1 (ICAM-1) [21]. Together with ox-LDL, activated leukocytes migrate through the injured endothelial layer to the intimal layer of the vessel wall, where they produce inflammatory mediators which sustain the inflammatory process [6]. 

Likewise, arterial SMC respond to chemoattractants produced by activated leukocytes, such as platelet-derived growth factor, which results in SMC migration from the tunica media of the artery wall to the intimal layer. There, SMC produce collagen, elastin, and glycosaminoglycans, which are the principal components of mature atherosclerotic plaques [19].

T-lymphocytes play a dual role in atherosclerosis. Regulatory T-lymphocytes (Tregs) which produce transforming growth factor β (TGF-β) and IL-10, as well as Th-17 lymphocytes that secrete IL-17, have a protective effect, promoting stabilization of atherosclerotic plaques due to the expansion of the fibrous cap [22]. In contrast, Th1 lymphocytes promote atherosclerosis by exuding proinflammatory components such as interferon-γ and TNF-α [23].

Platelets have an underappreciated role in atherogenesis. One of the mechanisms of their contribution is through their interactions with endothelial cells which result in the release of chemokines and expression of adhesion molecules (E-selectin, P-selectin, VCAM-1, ICAM-1). This is then followed by IL-1β secretion leading to the recruitment of immune cells [24]. Furthermore, platelets are able to participate in the oxidization of LDL and to promote LDL uptake by macrophages, thus contributing to the development of atherosclerosis [25].

The nucleotide-binding oligomerization domain-like receptors, pyrin domain-containing 3 (NLRP3) inflammasome, which is a component of the innate immune system, has been linked to several inflammatory disorders such as periodic cryopyrin syndromes, Alzheimer’s disease, diabetes, atherosclerosis, and more recently atrial fibrillation [26,27]. The NLRP3 inflammasome is responsible for caspase-1 activation and the secretion of proinflammatory cytokines IL-1β and IL-18 during microbial infection and cellular damage thus can contribute to ASCVD development [26]. 

Cells involved in the development of atherosclerosis secrete EVs and miRNAs that modulate inflammation of the blood vessel wall [18]. EVs are lipid membrane particles that are naturally released from cells into blood and other body fluids, which transport biomolecules between cells [28]. Multiple EVs subtypes are associated with pathophysiological processes underlying atherosclerosis and CVD, such as endothelial dysfunction, accumulation of lipids in the vessel wall, migration of immune cells, calcification of atherosclerotic plaques, and plaque rupture [29]. Moreover, EVs are involved in the adverse myocardial remodeling following acute myocardial infarction (AMI), making them not only biomarkers but also therapeutic targets in CVD [30,31]. The role of EVs in atherosclerotic development and progression has been summarized elsewhere [32].

MiRNAs are small, non-coding RNAs that regulate post-transcriptional gene expression [33]. For example, miR-181a-5p and its passenger strand miR-181a-3p act as negative post-transcriptional regulators of the NF-κB signaling pathway, thus contributing to inflammation. Restoring the expression of these miRNAs retarded not only the inflammation but also the progression of atherosclerosis in apoE−/− mice [34]. Additionally, miR-30c and miR-126p had a protective effect on ASCVD. MiR-30c was associated with hyperlipidemia and atherosclerosis reduction in the Western diet-fed mice, while miR-126p limited atherosclerosis by suppressing Notch1 inhibitor delta-like-1 homolog in Mir126−/− mice [35,36]. Conversely, miR-143 and miR-145 were characterized as vascular SMC physiology regulators, which are involved in SMC proliferation, migration, and apoptosis. miR-19b, miR-33, miR-92a, miR-144-3p, miR-155, and miR-342-5p were also shown to promote atherosclerosis [37,38,39,40,41,42]. Therefore, it is suggested that the dysregulation of miRNAs expression is associated with ASCVD development and progression [43]. Similar to EVs, miRNAs are also involved in post-infarction myocardial repair and could be used to design novel therapies to prevent and/ or reverse myocardial damage [44,45].

Discussing the role of inflammation in the development of atherosclerosis, good models of systemic inflammation are patients with Chronic Inflammatory Rheumatic diseases (CIRD) such as rheumatoid arthritis (RA) or systemic lupus erythematosus (SLE). Patients with CIRD are at increased risk of accelerated atherosclerosis compared with a healthy population [46]. Moreover, therapy with anti-inflammatory drugs such as anti-TNF or anti-IL-1 significantly reduces the risk, pointing to inflammation as the cause of the excessive risk of ASCVD [47].

Altogether, inflammatory processes involved in ASCVD are complex and require further investigation. Accumulating evidence shows that not only classical triggers of endothelial injury and inflammation but also infections may trigger and aggravate ASCVD. It remains unclear whether the mechanism by which infections affect atherosclerosis is similar to that in CIRD or whether additional factors play a role. There are also no direct comparisons between CIRD and infectious diseases in terms of ASCVD risk, thereby raising a need for additional research on this topic. However, the next paragraph focuses on infections as a risk for ASCVD. 

## 3. Infections and Atherosclerotic Cardiovascular Diseases

The mechanisms underlying the increased risk of ASCVD in course of infections and discussed in this paragraph are summarized in Figure 2.

### 3.1. Gastrointestinal Tract Infections

#### 3.1.1. Periodontal Disease

Periodontal disease is a common bacterial infection that is associated with some systemic disorders, including ASCVD [48]. Periodontal disease is mainly caused by *Aggregatibacter actinomycetemcomitans, Porphyromonas gingivalis,* and *Tannerella forsythia* [49]. A recent meta-analysis including 30 cohort studies demonstrated that there is a significant increase in the risk of ASCVD incidence in patients with periodontal disease, compared to those without (relative risk, RR = 1.20, 95% CI: 1.14–1.26) [50]. Among all types of ASCVD, the risk of stroke was highest (RR = 1.24, 95% CI: 1.12–1.38). The risk was higher in men, compared to women (RR = 1.16, 95% CI: 1.08–1.25) and higher in severe periodontal disease, compared to a mild stadium (RR = 1.25, 95% CI: 1.15–1.35). These results are consistent with previous studies that indicated elevated ASCVD prevalence in patients suffering from periodontal disease [51].

Periodontal pathogens can enter the circulation and cause bacteremia. Moreover, they have been found in atherosclerotic lesions, suggesting their direct impact on atherosclerosis [52]. Periodontal disease modifies traditional risk factors, since elevated serum levels of LDL, oxLDL, and triglycerides were described in such patients [53]. Negative aspects of periodontitis appeared reversible after treatment. The concentrations of pro-inflammatory molecules such as CRP, IL-1, IL-6, TNF-α, and fibrinogen are higher in patients with periodontitis and significantly reduced after the therapy [54]. Endothelial function measured by flow-mediated dilatation also improved after periodontal treatment both in patients without ASCVD and in patients after AMI [55,56]. Moreover, intensive periodontal treatment (non-surgical periodontal therapy with supra and subgingival instrumentation) caused a significantly larger decrease in systolic and diastolic blood pressure, compared to conventional treatment (supragingival debridement only) [57]. Periodontal diseases might affect prognosis after coronary events, and also in primary prevention, arterial hypertension and its consequences [58,59]. Altogether, early detection and proper treatment of periodontitis seem to be an important intervention not only to improve dental hygiene but also to reduce the risk of ASCVD development and progression [52]. 

#### 3.1.2. Helicobacter Pylori

*H. pylori* is the most common chronic bacterial infection in humans, with an estimated prevalence of 48.5% worldwide [60,61]. To date, numerous studies have shown the association between *H. pylori* infection and ASCVD. Recently, a meta-analysis involving 7522 cases and 8311 controls demonstrated that *H. pylori* infection increased the odds of an acute coronary syndrome (ACS) (odds ratio, OR = 2.03, 95% confidence interval, CI 1.66–2.47), especially in the developing countries, compared to the developed countries (OR = 2.58 vs. OR = 1.69, respectively) [62]. A suggested explanation for the discrepancy between the developing and developed countries is the access to primary prevention therapies against ASCVD (i.e., lipid-lowering drugs) which modifies the course of the disease [62]. Another meta-analysis (*n* = 19,691) indicated an increased risk of atherosclerotic cardiovascular events in patients with *H.pylori* infection (OR = 1.51, 95% CI 1.34–1.70), compared to those not infected, especially regarding AMI and cerebrovascular disease (OR = 1.80, OR = 1.54, respectively) [63]. Both meta-analyses pointed out that the strains of *H. pylori* with the cytotoxin-associated antigen A (CagA) significantly increase the risk of ASCVD (OR = 2.39 in Fang et al. [62], OR = 1.73 in Wang et al. [63]), suggesting that these more pathogenic strains may play a greater role in atherogenesis by inducing a more severe inflammatory response.

The role of *H. pylori* in the pathogenesis of ASCVD might be due to the systemic effects of the chronic inflammation, induced by *H. pylori* in the gastric mucosa [64]. Due to the changes in the gastric epithelium colonized by *H. pylori*, bacterial antigens may contact the immune cells via pathogen recognition receptors (PRRs) and induce the release of proinflammatory cytokines. Furthermore, the soluble antigens of *H. pylori* can enter the circulation and therefore induce a systemic immune response. *H. pylori* lipopolysaccharide (LPS) binds LPS-binding protein (LBP) subsequently forming a complex with LDL or ox-LDL which may explain the presence of LPS in atherosclerotic plaques [65]. The mere presence of LPS in the bloodstream already has a strong proinflammatory effect, as demonstrated in studies on sepsis of *Escherichia coli* etiology, so in this mechanism, other gastrointestinal bacteria, not only *H. pylori*, may exhibit proatherogenic effects if endotoxemia occurs [66]. Another proposed mechanism is molecular mimicry, where the heat shock protein 60 derived from *H. pylori* (Hp-HSP60) can cross-react with endogenous HSP60 and cause an immune response driven by Th1 lymphocytes, resulting in endothelium damage [67]. Systemic inflammation in the course of *H. pylori* infection is also reflected by increased concentrations of biomarkers of inflammation, such as CRP, platelet-to-lymphocyte ratio (PLR), and neutrophil-to- lymphocyte ratio (NLR), which were associated with the progression of ASCVD in patients without acute AMI who underwent coronarography [68,69]. Moreover, the association of *H. pylori* with traditional risk factors for ASCVD, such as smoking (smokers had increased risk of *H. pylori* infection), artery stiffness (increased pulse wave velocity in infected patients), increased HbA1c level or dyslipidemia (higher levels of total cholesterol, LDL and triglycerides, and lower HDL level in infected patients) were also described. However, since the risk (age, socioeconomic status, dietary patterns, cigarette smoking) and beneficial (aspirin usage, antibiotic treatments) factors of *H. pylori* infection and atherosclerosis often co-exist, it is not clear whether there is a causal relationship between both entities [9]. Nevertheless, evidence from large meta-analyses endorsed by pathomechanistic explanations suggests that *H. pylori* may contribute to ASCVD development and should be effectively treated.

#### 3.1.3. Hepatitis C Virus (HCV)

According to a recent meta-analysis including 341,739 subjects, *HCV* infection is associated with an increased risk of ASCVD (RR = 1.28, 95% CI: 1.18–1.39). The authors calculated that globally, 1.5 million disability-adjusted life-years (DALYs) per year were lost due to *HCV*-related ASCVD [70]. Similarly, another meta-analysis demonstrated that *HCV* infection was a risk factor for coronary artery disease (RR = 1.25; 95% CI: 1.12–1.40) [71]. Patients with chronic *HCV* infection had elevated concentrations of cardiac and inflammatory biomarkers, such as N-terminal pro-B-type natriuretic peptide (NT-proBNP), soluble ICAM-1, soluble VCAM-1, CRP, IL-6, and TNF-α compared to non-infected patients, suggesting cardiac failure, endothelial dysfunction, and pro-inflammatory state [72]. Furthermore, *HCV*-infected patients had lower levels of total cholesterol, high-density lipoprotein (HDL), LDL, and triglycerides, compared to non-infected, which indicates a pathomechanism unrelated to traditional risk factors [73].

Patients with *HCV* and *HIV* coinfection are a group of special interest because of the increased risk of ASCVD, compared to those with *HIV* mono-infection (pooled RR = 1.24; 95% CI: 1.07–1.40). As the *HCV* treatment with novel direct-acting antiviral agents (DAAs) was shown to reduce the rate of ASCVD events, it has been proposed to extend the screening for ASCVD in *HCV*-infected patients, and for *HCV* infection in ASCVD patients to improve the management of both diseases [74,75].

### 3.2. Respiratory Tract Infections

#### 3.2.1. Pneumonia

The association between bacterial pneumonia and the risk of ASCVD was observed. In a group of 591 patients hospitalized due to pneumonia, cardiovascular complications (including MI, stroke, and fatal coronary heart disease events) occurred in 34.85% of patients within 10 years of hospital discharge. The risk was highest (adjusted HR = 4.07 (95% CI: 2.86–5.27)) in the first 30 days after pneumonia and remained significantly higher up to 10 years after the infection (between 9 and 10 years after hospitalization, adjusted HR = 1.86 (95% CI: 1.18–2.55)) [76].

Pathogens related to pneumonia such as *Streptococcus pneumoniae*, *Chlamydophila pneumoniae*, and *Mycoplasma pneumoniae* induced progression and instability of atherosclerotic plaques in an animal model. They were shown to trigger both local endothelial inflammation and persistent systemic pro-inflammatory state, as reflected by elevated CRP, IL-6, and IL-18 after pneumonia [11]. *S. pneumoniae* may also directly invade the heart and lead to cardiomyocyte necroptosis (a programmed form of necrosis induced by inflammation), which further enhances the inflammatory response and may lead to long-term cardiac remodeling [77,78]. Pneumonia is also characterized as a prothrombotic state due to several pathogen-driven mechanisms, including increased platelet activation. Platelets express numerous immunoreceptors that enable them to recognize intravascular pathogens, including toll-like receptors (TLR) which bind their respective pathogen-associated molecular patterns (PAMPs) and damage-associated molecular patterns (DAMPs). For example, platelet TLR-2 can directly bind the antigens of *S. pneumoniae*, which leads to platelet activation, the first step in thrombus formation [79,80]. Platelets also bind immunoglobulins (Ig) or complement fragments, present in the bloodstream in course of infection. Thus, there is a direct pathophysiological link between bacterial pneumonia and thrombotic complications, such as AMI or stroke [11,81].

In addition to pneumonia itself, it has been postulated that antibiotics used to treat pneumonia (macrolides and fluoroquinolones) are associated with increased cardiovascular risk [82]. However, the risk associated with macrolide use was substantially attenuated after adjustment for clinical variables, and the risk associated with fluoroquinolones was no longer significant [83]. Concurrently, measures to prevent pneumonia such as polysaccharide pneumococcal vaccination (PPV23) decreased the risk of ASCVD, especially in individuals aged >65 years, supporting the vaccination benefits among the elderly population at high risk of pneumonia [84]. 

#### 3.2.2. Cytomegalovirus (*CMV*)

Although the association between *CMV* infection and ASCVD development has been controversial, recent meta-analyses supported this association [10]. A meta-analysis including 34,564 participants (4789 ASCVD patients) demonstrated that *CMV* infection increased the relative risk of ASCVD by 22% (RR = 1.22, 95% CI: 1.07–1.38, *p* = 0.002) [85]. In the group of 12,903 cases and 16,097 controls, patients exposed to *CMV*, assessed on the IgG tests, had a higher risk of ASCVD (OR = 1.70; 95% CI: 1.43–2.03) [86]. Interestingly, patients in the active phase of the disease, assessed based on the IgM and polymerase chain reaction (PCR) tests, had an even higher risk of ASCVD (OR = 2.88, 95% CI: 1.87–4.43; OR = 2.56, 95% CI: 1.46–4.49, respectively), suggesting that active infection may directly contribute to ASCVD events.

*CMV* infection is associated with a systemic proinflammatory state of the host, reflected by higher levels of pro-inflammatory molecules contributing to ASCVD, such as CRP, IL-1β, IL-6, TNF-α, and IFN-γ [87,88]. The increased ASCVD risk in course of *CMV* infection is likely associated with the negative effect of *CMV* on endothelial function [10]. *CMV* can directly infect endothelial cells and SMC, dysregulating their normal activity and triggering a local inflammatory response [89]. Further, *CMV* indirectly affects endothelial function by upregulating the production of asymmetric dimethylarginine (ADMA), which is the endogenous endothelial nitric oxide synthase (eNOS) inhibitor, thus decreasing the concentration of the vasoprotective NO [90].

#### 3.2.3. Severe Acute Respiratory Syndrome Coronavirus-2 (SARS-CoV-2)

*SARS-CoV-2*, responsible for coronavirus disease-2019 (COVID-19) is the most recent pathogen described to be associated with CVD complications, including ASCVD, which significantly increase mortality. These complications include viral myocarditis, classic myocardial infarction due to infection-induced atherosclerotic plaque instability, stress-induced cardiomyopathy (Takotsubo syndrome), venous thromboembolism, heart failure, and arrhythmias [91,92,93].

In a meta-analysis of 12 studies including 3044 confirmed COVID-19 cases, the most common cardiovascular complication of COVID-19 were myocardial injury (21.2%, 95% CI: 12.3–30.0%), arrhythmia (15.3%, 95% CI: 8.4–22.3%), and heart failure (14.4%, 95% CI: 5.7–23.1%) [94]. In COVID-19 patients, cardiac injury was associated with higher risk of mortality (RR = 7.79; 95% CI: 4.69–13.01), ICU admission (RR = 4.06; 95% CI: 1.50–10.97), mechanical ventilation (RR = 5.53; 95% CI: 3.09–9.91), and developing coagulopathy (RR = 3.86; 95% CI: 2.81–5.32; I^2^ = 0%).

The mechanisms potentially explaining the association between COVID-19 and CVD include cytokine storm and thrombotic microangiopathy [95,96]. Severe COVID-19 is associated with a cytokine storm, reflected by elevated concentrations of TNF-α, IL-1β, IL-2, IL-6, CRP, and ferritin [97]. Reports of elevated levels of proinflammatory cytokines in COVID-19 have resulted in many studies on the efficacy of interleukin inhibitors for the treatment of COVID-19 and some of them (anakinra, tocilizumab, sarilumab) have demonstrated efficacy in several subgroups [98]. A cytokine storm leads to massive endothelial injury and immunothrombosis, contributing to multi-organ failure [99]. *SARS-CoV-2* may directly invade endothelial cells and cardiac cells (mostly interstitial fibroblasts), causing myocarditis and endothelitis. The state of hypercoagulability associated with COVID-19 is reflected by elevated concentrations of D-dimer and other fibrin degradation products [100,101]. In addition, medications used to combat COVID-19 (e.g., remdesivir, lopinavir, chloroquine, methylprednisolone) may have a negative impact on CVD, compounding the risk [102].

Importantly, even asymptomatic *SARS-CoV-2* infection may have cardiovascular complications, termed post-COVID-19 heart syndrome. Persistent cardiac involvement and ongoing myocarditis were observed in COVID-19 convalescents (78% and 60%, respectively) in cardiac magnetic resonance even 3 months after the acute phase of the disease. Moreover, imaging changes have been described in people who had no symptoms of COVID-19, raising the possibility that post-COVID heart syndrome may be the first sign of a past infection [103]. Hence, whereas the short-term complications of acute COVID-19 are relatively clear, the knowledge regarding long-term effects, especially on the progression of ASCVD, requires further investigation.

### 3.3. Immune System Infections

#### Human Immunodeficiency Virus (*HIV*)

In *HIV*-positive individuals, the rate of ASCVD is two-fold higher, compared to the general population, making ASCVD the main cause of morbidity and mortality among *HIV*-positive patients [12]. The increased risk persists in patients with well-controlled viral load [104].

*HIV*-related mechanisms underlying ASCVD include proinflammatory effects of *HIV* proteins, CD4^+^ T-cell depletion, increased intestinal permeability, and altered cholesterol metabolism [105]. The depletion of CD4^+^ T-cell in the intestines in acute *HIV* infection enables the translocation of microbial products, such as lipopolysaccharide (LPS) derived from gut bacteria to bloodstream and results in increased production of TNF-α, IL-6, IL-8, and ox-LDL, proinflammatory factors related to ASCVD. Elevated levels of LPS persist after the initiation of antiretroviral treatment (ART) [12].

ART may be also responsible for the excessive risk of ASCVD in *HIV*-positive patients. Older drugs, such as abacavir, lopinavir, and ritonavir, increase the risk of CVD development due to their side effects, including altered glucose and lipid metabolism, impaired left ventricular function, or cardiac myopathy due to mitochondrial toxicity [105]. Furthermore, *HIV* infection is associated with a higher prevalence of co-infection with *CMV*, *HCV*, and periodontal disease, which may additionally contribute to the increased ASCVD risk [12]. As the lifespan expectancy of *HIV*-positive patients raised due to the highly efficient ART, it is important to appropriately manage both the traditional risk factors of ASCVD, as well as to intensively treat co-infections in this especially vulnerable patients’ population.

### 3.4. Dermatologic Infections 

#### Herpes Simplex Virus (HSV)

The association between *HSV* and atherosclerosis has been described already in the ‘90s, although it has remained controversial [106]. On one hand, it was demonstrated that subjects exposed to both *HSV-1* and *HSV-2* infection had an increased risk of atherosclerosis (OR = 1.77, 95% CI: 1.40–2.23, *p* < 0.0001; OR = 1.37, 95% CI: 1.13–1.67, *p* < 0.005, respectively) [13]. Additionally, *HSV* infection may promote periodontal disease, and exposure to these two conditions increases the risk of ASCVD. Moreover, patients infected with *HSV* have significantly decreased HDL levels [107]. However, another study that enrolled 14,415 participants showed that only *HSV-2*, but not *HSV-1*, was associated with premature ASCVD (before the age of 50) (OR = 1.56, 95% CI:1.09–2.21, *p* = 0.014 for *HSV-2*) [108]. The authors observed that the level of CRP was higher in patients exposed to *HSV-2* than in those exposed to *HSV-1*, suggesting a stronger pro-inflammatory effect of *HSV-2*. Both types of *HSV* can be found in atherosclerotic plaques, but also in a normal artery. Hence, the clinical relevance of this finding is unknown [109].

## 4. Therapeutic Implications

Regarding the excessive risk of ASCVD related to infections, it is crucial to establish the effective prevention and treatment of both entities. Here, we focus on the currently available strategies. However, since the residual risk related to chronic infections remains an important cardiovascular risk factor, there is an ongoing search for new therapeutics with the anti-inflammatory and/or anti-infective modes of action. This is particularly important given the strong literature reports of the ineffectiveness of anti-infective drugs, including antibiotics in the prevention of ASCVD [110]. The anti-inflammatory and/or anti-infective mechanisms of currently available drugs that may affect residual risk are shown in Figure 3. A summary of clinical studies evaluating the cardiovascular outcomes in patients receiving anti-inflammatory drugs is shown in Table 1.

### 4.1. Statins

Numerous studies indicate the pleiotropic effects of statins. Besides lowering the LDL concentration, statins also affect other pathways related to atherosclerosis, that is increase NO production, decrease ICAM-1 expression, and decrease CRP level. Both lowering the LDL and other mechanisms of action are responsible for the anti-inflammatory effects of statins [120]. Ridker et al. [111] in their study on 15,548 initially healthy men and women demonstrated that during rosuvastatin therapy (20 mg daily), there is a need to achieve a therapeutic target both in LDL (<70 mg/dL) and high-sensitivity C-reactive protein (hsCRP) (<2 mg/L) level to improve outcomes, compared to one target only, or none of them. Moreover, the prognosis was better, when the hsCRP concentration was below 1 mg/L, indicating the combined lipid-lowering and anti-inflammatory role of statins during therapy. The overall number of patients who met both conditions (LDL < 70 mg/dL and hsCRP < 2 mg/L) was 2685 (of 15,548) and the number of patients who met more intensified criteria (LDL < 70 mg/dL and hsCRP < 1 mg/L) was only 944 [111]. Intriguingly, research on proprotein convertase subtilisin/kexin type 9 (PCSK9) inhibitors showed that the decrease in CRP level is not inevitably associated with LDL reduction. According to the posthoc analysis of the SPIRE trials (9738 patients receiving both statins and bococizumab (PCSK9 inhibitor)), there is evidence of residual inflammatory risk among patients treated with both statin and PCSK9 inhibitors [121]. It has been hypothesized that statins may have an additional mechanism of action, which may be more potent than in the case of PCSK9 inhibitors [122].

Statins, because of their pleiotropic effects, have also been evaluated in the treatment of sepsis in adult patients. However, the meta-analysis on the group of 2628 patients with sepsis resulted in no significant reduction of 30-day all-cause mortality in patients receiving statins, compared to no treatment or placebo [123].

### 4.2. P2Y12 Inhibitors

Whereas the role of P2Y12 inhibitors in secondary prevention of thrombotic events is well-established, their anti-inflammatory and anti-infective effect is a relatively new topic. Both ticagrelor and clopidogrel decrease the exposure of P-selectin, and concentrations of soluble P-selectin and CD40L to a similar extent in patients with AMI [124,125]. In the inflammation induced by bacterial endotoxin, both ticagrelor and clopidogrel were associated with the reduction of systemic inflammation, reflected by platelet-monocyte aggregate formation, TNF-α, and IL-6 levels. Moreover, ticagrelor, but not clopidogrel, significantly lowered peak levels of IL-8, increased the concentration of anti-inflammatory cytokine IL-10, and additionally altered leukocyte trafficking and suppressed prothrombotic changes in fibrin clot ultrastructure [112,126]. Ticagrelor was also associated with the decreased concentration of proinflammatory and prothrombotic EVs after AMI, compared to clopidogrel [127]. This effect could be explained by a dual antiplatelet activity of ticagrelor–blocking P2Y12 receptor and increasing the extracellular concentration of adenosine–which might contribute to the enhanced anti-inflammatory effect [128]. Whether other P2Y12 inhibitors, including prasugrel and cangrelor, have similar effects remains to be established. 

### 4.3. Angiotensin-Converting Enzyme Inhibitors (ACE-I) and Angiotensin Receptor Blockers (ARBs)

Angiotensin II is an important component of the renin-angiotensin system (RAS) and is associated with the promotion of atherosclerosis [129]. Treatment with either ACE-I or ARBs is effective not only in course of atherosclerosis but for a wide variety of inflammatory diseases such as arthritis, steatohepatitis, colitis, pancreatitis, and nephritis [130]. Therapy with ACE-I ramipril in a daily dose of 10 mg reduced the hsCRP concentration in patients with atherosclerosis (−32% after 3 months; *p* = 0.0002), indicating a potential beneficial effect on systemic inflammation [131]. In another study, administration of perindopril lowered TNF-α and D-dimer but did not affect the concentration of CRP and fibrinogen [132]. In the study including 14,703 patients with ASCVD, the rate of atherosclerotic events was reduced to a similar extent by treatment with captopril or valsartan [113]. Interestingly, atheroma volume was significantly reduced when ramipril was added to rosuvastatin therapy, compared to rosuvastatin only, suggesting the synergistic anti-inflammatory effects of combined therapy [133]. 

### 4.4. Colchicine

Colchicine is an anti-inflammatory drug that blocks the NLRP3 inflammasome, resulting in lowered IL-1β and IL-18 production, and subsequently IL-6 and CRP levels [134]. Although the CRP level in CAD patients was decreased after the colchicine administration, the endothelial function measured by flow-mediated dilatation was not improved [135]. Colchicine reduced the local coronary production of pro-inflammatory cytokines and chemokines in patients after acute coronary syndromes [134,136]. In the recent randomized clinical trial including 4745 patients shortly after AMI, a daily administration of 0.5 mg of colchicine was associated with a decreased risk of ischemic cardiovascular events, compared with placebo (HR = 0.77; 95% CI: 0.61–0.96; *p* = 0.02) [114]. A similar trial on the group of 5522 patients with the chronic coronary disease showed a 31% reduction in the risk of cardiovascular events (HR = 0.69; 95% CI:0.57–0.83; *p* < 0.001) in patients receiving 0.5 mg colchicine daily, compared with the placebo group [115]. Colchicine is a well-known, inexpensive drug with little potential for serious adverse effects, which has the potential to reduce the further residual risk in ASCVD patients receiving a standard therapy [137]. However, despite its low potential for side effects, they can be a problem in long-term treatment with colchicine. In literature, the most common are diarrhea, nausea, and flatulence but there is also a risk of drug interactions with statins that increase the risk of myopathy [114,138]. In clinical practice, it should be always considered whether the benefits outweigh the risk and eventually the particular population of patients that could benefit from the therapy should be identified. 

### 4.5. Anti-Cytokine Drugs

Therapies aiming at reducing the risk of ASCVD with anti-cytokine drugs have the potential to revolutionize the treatment of any disease with inflammatory background, including ASCVD, cancer, and autoimmune diseases. Hence, many studies have been conducted in this field. However, prior research was based mostly on patients with autoimmune diseases like rheumatoid arthritis, which limits the applicability of the results [120]. For example, TNF antagonists reduced the risk of cardiovascular events, compared to standard therapy with disease-modifying anti-rheumatic drugs in patients with rheumatoid arthritis [118]. However, recently Ridker et al. performed the first randomized controlled trial to evaluate an impact of a human anti-IL-1β monoclonal antibody—canakinumab—on recurrent cardiovascular events in patients after AMI with a hsCRP level of 2 mg/L or more [117]. Canakinumab reduced the rate of recurrent atherosclerotic cardiovascular events, compared with placebo (HR = 0.85, 95%CI: 0.74–0.98; *p* = 0.021), independent of lipid-level lowering. However, the therapy was associated with an increased rate of fatal infections and there was no significant difference in all-cause mortality. Nevertheless, this trial was the first to directly confirm the role of inflammation in the pathogenesis of ASCVD in a clinically relevant model. So far, the evidence considering the clinical benefits and cost-efficacy of canakinumab is insufficient to recommend the routine use of anti-cytokine drugs in addition to standard therapy to lower the ASCVD risk [139].

### 4.6. Methotrexate

Methotrexate is an immune-system suppressive agent widely used in inflammatory diseases such as rheumatoid arthritis. Low-dose therapy with methotrexate was hypothesized to be beneficial in ASCVD because of its anti-inflammatory effect. However, a large study including 4786 patients with previous MI or coronary disease who had additional type 2 diabetes or metabolic syndrome showed no difference in atherosclerotic cardiovascular events between patients on low-dose methotrexate therapy (15–20 mg per week), compared to placebo [119]. Besides, methotrexate caused elevations in the liver enzyme levels, hematological adverse effects, and a higher incidence of non-basal-cell skin cancers. It suggests that the effectiveness of methotrexate in the reduction of atherosclerotic cardiovascular events rates may be limited to conditions with a high level of inflammation such as rheumatoid arthritis, but without affecting the residual risk [119]. 

## 5. Conclusions

Several infections including *H. pylori* infection, periodontal disease, and viral infections are associated with the increased risk of ASCVD, both directly by damage to the heart muscle and vasculature, and indirectly by triggering a systemic proinflammatory state. Hence, beyond the optimal management of the traditional ASCVD risk factors, infections should be considered as an important non-classical cardiovascular risk factor to enable early diagnosis and appropriate treatment. Numerous drugs routinely administered in patients with ASCVD including statins, P2Y12 inhibitors, and ACEi/ARBs were shown to have anti-inflammatory and/or anti-infective properties. However, no randomized controlled trials have hitherto evaluated their efficacy in patients with infective diseases. Novel strategies aiming at residual risk reduction have been proposed, including colchicine, anti-cytokine drugs, and methotrexate. However, more evidence regarding the risk-to-benefit ratio and cost-efficiency of these drugs is required before their clinical routine implementation in patients with infection-related high residual risk of CVD. Additional research is also needed on new therapeutic strategies, such as, for example, anti-atherosclerotic vaccines that already exist in the literature and where the biggest challenge is finding the appropriate antigen to immunize [110]. Nevertheless, further progress in understanding and therapeutic application of the relationship between infections/chronic inflammation and ASCVD requires well-designed, large studies with long-term follow-up. 

## Figures and Tables

**Figure 1 jcm-10-02539-f001:**
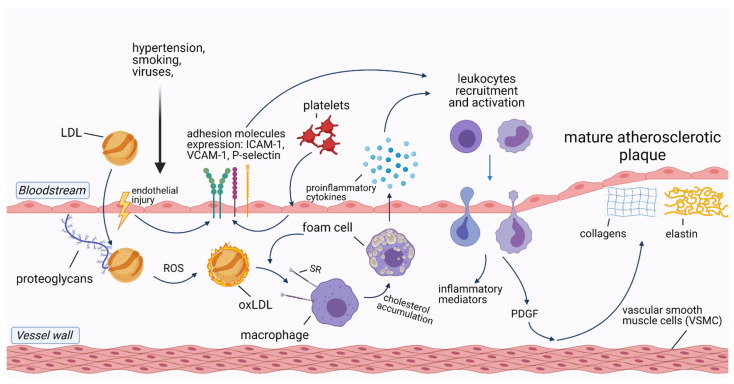
Pathophysiology of atherosclerotic plaque formation. Abbreviations: ICAM-1—intercellular adhesion molecule-1; LDL—low-density lipoprotein; oxLDL—oxidized LDL; PDGF—platelet-derived growth factor; ROS—reactive oxygen species; SR—scavenger receptor; VCAM-1—vascular cell adhesion protein 1. Created with Biorender.com, licensed version.

**Figure 2 jcm-10-02539-f002:**
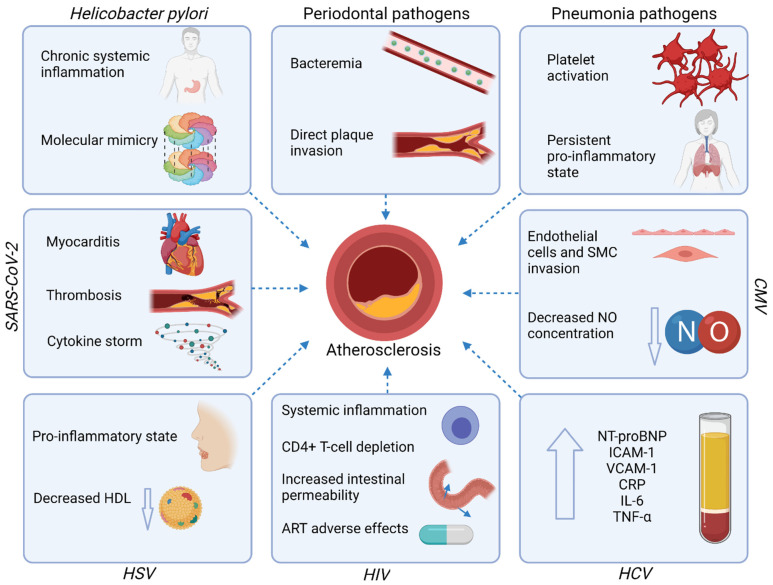
Mechanisms underlying the increased risk of ASCVD in course of infections. Abbreviations: ART—antiretroviral therapy; CMV—Cytomegalovirus; CRP—C-reactive protein; HCV—Hepatitis-C virus; HDL—high-density lipoprotein; HIV—Human immunodeficiency virus; HSV—Herpes simplex virus; ICAM-1—Intercellular adhesion molecule-1; IL—interleukin; NO—nitric oxide; NT-proBNP—N-terminal pro-B-type natriuretic peptide; SARS-CoV-2—Severe acute respiratory syndrome coronavirus-2; SMC—Smooth muscle cells; TNF-α—tumor necrosis factor-alpha; VCAM-1—Vascular cell adhesion protein 1. Created with Biorender.com, licensed version.

**Figure 3 jcm-10-02539-f003:**
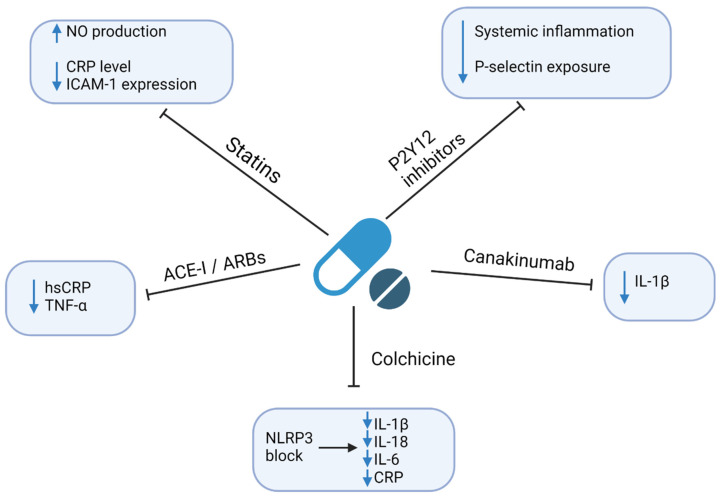
Anti-inflammatory and/or anti-infective mechanisms of currently available drugs. Abbreviations: ACEI—angiotensin-converting enzyme inhibitor; ARBs—angiotensin II receptor blocker; CRP—C-reactive protein; hsCRP—high-sensitivity CRP; ICAM-1—Intercellular Adhesion Molecule 1; IL—interleukin; NLRP3—nucleotide-binding oligomerization domain-like receptors, pyrin domain-containing 3; NO—nitric oxide; TNF-α—tumor necrosis factor-alpha. Created with Biorender.com, licensed version.

**Table 1 jcm-10-02539-t001:** Summary of clinical studies evaluating the cardiovascular outcomes in patients receiving anti-inflammatory drugs.

Authors	Therapy	Mechanism of Action	Information about Study	Outcomes	Effect
Ridker et al., 2009 [111]	Rosuvastatin vs. placebo	HMG-CoA inhibitor, pleiotropic effects	A randomized, double-blind, placebo-controlled trial including 15,548 initially healthy men and women	Cardiovascular death, non-fatal stroke, non-fatal AMI, hospitalization due to unstable angina, revascularization	↓ risk of adverse outcomes (HR = 0.35; 95% CI: 0.23–0.54; *p* < 0.0001)
Thomas et al., 2015 [112]	Ticagrelor vs. clopidogrel vs. placebo	Inhibition of P2Y12 receptor	Randomized injection of *E. coli* endotoxins to 30 healthy volunteers (10-ticagrelor, 10-clopidogrel, 10-placeboes)	Concentrations of inflammatory biomarkers	Ticagrelor and clopidogrel: ↓ IL6, TNF-α, CCL2 Only ticagrelor: ↓ G-CSF, IL-8; ↑ IL-10; ↔ hsCRP
McMurray et al., 2006 [113]	Valsartan vs. captopril	ARB or ACE inhibition	Randomized 14,703 high-risk patients with acute MI to receive captopril or valsartan or the combination of the two	All-cause mortality, cardiovascular mortality, non-fatal cardiovascular events	↓ risk of adverse outcomes; similar effect of ARBs and ACE-I (HR = 0.97; 95% CI:0.91–1.03; *p* = 0.286)
Tardif et al., 2020 [114]	Colchicine 0.5 mg daily vs. placebo	NLRP3 inflammasome inhibitor	A randomized, double-blind, placebo-controlled trial including 4745 patients with recent AMI (~2 weeks before)	Cardiovascular death, resuscitated cardiac arrest, AMI, stroke, coronary revascularization	↓ risk of adverse outcomes (HR = 0.77; 95% CI: 0.61–0.96; *p* = 0.02)
Nidorf et al., 2019 [115]	Colchicine 0.5 mg daily vs. placebo	NLRP3 inflammasome inhibitor	A randomized, placebo-controlled, double-blind trial including 5522 patients with chronic coronary syndrome	Cardiovascular death, MI, ischemic stroke, coronary revascularization	↓ risk of adverse outcomes (HR = 0.69; 95% CI: 0.57–0.83; *p* < 0.001)
Nidorf et al., 2013 [116]	Colchicine 0.5 mg daily vs. placebo	NLRP3 inflammasome inhibitor	A prospective, randomized, observer-blinded, placebo-controlled clinical trial including 532 patients with stable coronary disease	Acute coronary syndrome, out-of-hospital cardiac arrest, ischemic stroke	↓ risk of adverse outcomes (HR = 0.33; 95% CI: 0.18–0.59; *p* < 0.001)
Ridker et al., 2017 [117]	Canakinumab 150 mg every 3 months vs. placebo	Monoclonal anti-IL-1β antibody	A randomized, double-blind, placebo-controlled trial including 10,061 patients with previous AMI and hsCRP ≥ 2 mg/L	Non-fatal myocardial infarction, nonfatal stroke, cardiovascular death	↓ risk of adverse outcomes HR = 0.85 (95% CI: 0.74–0.98; *p* = 0.021)
Greenberg et al., 2010 [118]	TNF-α antagonists vs. DMARDs	TNF-α inhibition	A longitudinal cohort study of 10,156 rheumatoid arthritis patients enrolled in the US-based CORRONA database	Non-fatal MI, transient ischemic attack, stroke, cardiovascular death	↓ risk of adverse outcomes by TNF-α (HR = 0.39; 95% CI 0.19–0.82)
Ridker et al., 2019 [119]	Methotrexate 15–20 mg/week vs. placebo	Antimetabolite, immune-system suppressant	A randomized, double-blind, placebo-controlled trial including 4786 patients with previous MI or multivessel coronary disease, additionally with type 2 diabetes or metabolic syndrome	Nonfatal MI, nonfatal stroke, cardiovascular death, unstable angina	↔ adverse outcomes (HR = 0.96; 95% CI: 0.79–1.16; *p* = 0.67) ↔ hsCRP, IL-1β, IL-6 ↑ ALT, AST

Abbreviations: ACE—angiotensin-converting enzyme; ALT—Alanine transaminase; AMI—acute myocardial infarction; ARB—angiotensin II receptor blocker; AST—aspartate transaminase; CCL2—(C-C motif) ligand 2; CI—confidence interval; DMARDs—Disease-modifying antirheumatic drugs; *E.coli*—*Escherichia coli;* G-CSF—Granulocyte colony-stimulating factor; HMG-CoA—3-hydroxy-3-methyl-glutaryl-coenzyme A; HR—hazard ratio; hsCRP—high-sensitivity C-reactive protein; IL—interleukin; MI—myocardial infarction; NLRP3—nucleotide-binding oligomerization domain-like receptors, pyrin domain-containing 3; TNF-α—tumor necrosis factor-alpha.

## Data Availability

Not applicable.

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
