# Peer review of "Infections as Novel Risk Factors of Atherosclerotic Cardiovascular Diseases: Pathophysiological Links and Therapeutic Implications"

_jcm, 2021, doi:10.3390/jcm10122539_

Round 1

Reviewer 1 Report

In this review, Dr. Szwed and colleagues discussed the link between infection and cardiovascular diseases (CVD). Overall, this is a nice review article with a lot of insights, however I detected some limitations that could lower the significance / impact of the review. Moreover, I have some suggestions and comments to be addressed by the authors:

  • The current version of the manuscript is incomplete. The authors mentioned in the title about CVD but there is only a discussion about vessel diseases (CAD and atherosclerosis). Please add a detailed information about the implications of inflammation and infection on the heart (e.g., arrhythmias, cardiomyopathy, valves and heart failure).
  • The first paragraph of the introduction is incomplete. Please add some information about heart diseases (as I mentioned before) to balance with the vascular topic.
  • In Section 2, although CAD and atherosclerosis are good classical examples of the effect of inflammation on CV system, the authors need to add some more recent information about inflammation and  heart diseases, such as arrhythmias, structural and valvular heart diseases. The authors can get the information from these articles and other previous publications (Heijman et al. PMID: 32762493; Tschope et al. PMID: 33046850; etc.)
  • In organizing the subsection, please group the infections, bacteria and viruses per system, e.g., cardiorespiratory, gastrointestinal, genitourinary infections, etc. At the moment, everything is scattered and unorganized. This is important because these viruses are only some examples among many infectious agents out there which could affect CV system. 
  • Please cite this similar review from Pothineni et al. (PMID: 29020241).
  • The authors can benefit from the review by Hertanto et al. (https://www.preprints.org/manuscript/202104.0022/v1), in which they discussed hyperinflammation and potential benefits of anti-cytokine drugs in COVID-19.
  • I think NLRP3 inflammasome needs to be briefly discussed in section 2 about inflammation and CVD since it is currently an emerging topic with broad implications on CV system (Heijman et al. PMID: 32762493)
  • In addition to H.pylori, please add the implication of other GI infections, such as bacterial endotoxin (E.coli) as previously discussed by Sutanto and Lyon (PMID: 33873248). 
  • There is a typo in the affiliation number 1. Please fix.
  • Please (re)check for some grammatical errors and typos in the manuscript, for example:
    • line 11: CVD is ... or CVDs are ...
    • line 11: remove one of the "cause of"
    • line 33: "for a better"
    • line 37: "to oxidative stressors leads to the activation"

Author Response

Dear Reviewer,

we are thankful for the time and effort that you spent to provide an in-depth review of our review article. We corrected our manuscript according to your suggestions. Our response and corrections are listed below.

  1. In this review, Dr. Szwed and colleagues discussed the link between infection and cardiovascular diseases (CVD). Overall, this is a nice review article with a lot of insights, however I detected some limitations that could lower the significance / impact of the review.

We thank the Reviewer for appreciating our work.

  1. The current version of the manuscript is incomplete. The authors mentioned in the title about CVD but there is only a discussion about vessel diseases (CAD and atherosclerosis). Please add a detailed information about the implications of inflammation and infection on the heart (e.g., arrhythmias, cardiomyopathy, valves and heart failure).

 The first paragraph of the introduction is incomplete. Please add some information about heart diseases (as I mentioned before) to balance with the vascular topic.

 In Section 2, although CAD and atherosclerosis are good classical examples of the effect of inflammation on CV system, the authors need to add some more recent information about inflammation and  heart diseases, such as arrhythmias, structural and valvular heart diseases. The authors can get the information from these articles and other previous publications (Heijman et al. PMID: 32762493; Tschope et al. PMID: 33046850; etc.)

We thank the Reviewer for these comments. Indeed, we agree that a comprehensive discussion of the complications of infection in the context of a larger spectrum of different cardiovascular diseases could greatly enrich our work. However, it seems to us that this could significantly expand the volume of the article making it more difficult for the reader to assimilate. Therefore, we believe that for the sake of clarity and quality of the paper, we should focus on atherosclerotic cardiovascular diseases, taking the Reviewer's comment to heart while thinking about another paper regarding non-atherosclerotic complications of infections. Accordingly, we have modified our paper to avoid misleading the reader by specifying in the text and the title that the cardiovascular diseases we discussed belong to the group associated with atherosclerosis.

  1. In organizing the subsection, please group the infections, bacteria and viruses per system, e.g., cardiorespiratory, gastrointestinal, genitourinary infections, etc. At the moment, everything is scattered and unorganized. This is important because these viruses are only some examples among many infectious agents out there which could affect CV system. 

We thank the Reviewer for this suggestion. We have restructured our work according to the recommendations.

  1. Please cite this similar review from Pothineni et al. (PMID: 29020241).

We thank the Reviewer for this recommendation. Based on this position, we added a sentence in the introduction of Section 4 and Conclusions, as follows:  This is particularly important given the strong literature reports of the ineffectiveness of anti-infective drugs, including antibiotics in the prevention of ASCVD. Additional research is also needed on new therapeutic strategies, such as anti-atherosclerotic vaccines that already exist in the literature and where the biggest challenge is finding the appropriate antigen to immunize.

5.The authors can benefit from the review by Hertanto et al. (https://www.preprints.org/manuscript/202104.0022/v1), in which they discussed hyperinflammation and potential benefits of anti-cytokine drugs in COVID-19.

We thank the Reviewer for this suggestion. We added a sentence in 3.2.3. Section, based on proposed article, as follows: Reports of elevated levels of proinflammatory cytokines in COVID-19 have resulted in many studies on the efficacy of interleukin inhibitors for the treatment of COVID-19 and some of them (anakinra, tocilizumab, sarilumab) have demonstrated efficacy in several subgroups.

  1. I think NLRP3 inflammasome needs to be briefly discussed in section 2 about inflammation and CVD since it is currently an emerging topic with broad implications on CV system (Heijman et al. PMID: 32762493)

We thank the Reviewer for this comment. We added a paragraph about NLRP3 in Section 2, as follows: The NLRP3 inflammasome, which is a component of the innate immune system, has been linked to several inflammatory disorders such as periodic cryopyrin syndromes, Alzheimer's disease, diabetes, atherosclerosis, and more recently atrial fibrillation [26, 27]. NLRP3 inflammasome is responsible for caspase-1 activation and the secretion of proinflammatory cytokines IL-1β and IL-18 during microbial infection and cellular damage thus can contribute to ASCVD development.

  1. In addition to H.pylori, please add the implication of other GI infections, such as bacterial endotoxin (E.coli)as previously discussed by Sutanto and Lyon (PMID: 33873248). 

We thank the Reviewer for this suggestion. We added an implication about bacterial endotoxins in Section 3.1.1., based on this interesting, proposed article, as follows: The mere presence of LPS in the bloodstream already has a strong proinflammatory effect, as demonstrated in studies on sepsis of E.coli etiology, so in this mechanism, other gastrointestinal bacteria, not only H.pylori, may exhibit proatherogenic effects if endotoxemia occurs.

  1. There is a typo in the affiliation number 1. Please fix.

We corrected the typos.

  1. Please (re)check for some grammatical errors and typos in the manuscript, for example:
    • line 11: CVD is ... or CVDs are ...
    • line 11: remove one of the "cause of"
    • line 33: "for a better"
    • line 37: "to oxidative stressors leads to the activation"

We thank the Reviewer for this insights, we checked the text again and corrected the errors.

Altogether, we are grateful for the in-depth revision of our manuscript and we hope that it will be considered for publication.

On behalf of all Authors,

Sincerely,

Piotr Szwed

Reviewer 2 Report

The authors submitted a review about the role of infections as a risk factor for cardiovascular disease and summarized previously published results on the influence of various infections such as HIV, HSV and up-to-date SARS-CoV-2, as well as different anti-inflammatory or CV drugs, on cardiovascular risk and disease.

I would like to commend the authors for the preparation of this extensive work. I have the following comments:

General comments:

  • Systematic reviews should follow PRISMA guidelines (see http://prisma-statement.org/) While some aspects of PRISMA are fulfilled in the manuscript at hand, others are not (e.g. how were the reported studies identified,..). I recommend revisiting the PRISMA (abstract) checklist and adapt the manuscript accordingly.
  • I would like to encourage the authors to expand on the discussion/conclusion and outlook regarding this topic. E.g. what should future research focus on? Where do you think lie the biggest pitfalls or potential benefits?
    • Optional: shortly discussing the role of non-infectious chronic inflammatory disease in cardiovascular risk/disease might provide interesting additional insights (pathogen vs inflammation as cause for increased CV risk).
  • Formal revision of the language of the draft might further improve the quality of the manuscript.

Specific comments:

  • Table 1 could benefit from more detailed information about the study to better inform about the quality of the studies, e.g. type of study, multicenter vs single-center, country of conduct.
  • I would disagree with the statement made in line 462 that Colchicine has little potential for adverse effects, especially in a vulnerable patient population often exhibiting chronic and/or acute renal failure.
  • I think there is a reference missing in line 492.

Limitation of this review:

  • As a clinician and clinical researcher I cannot comment on the discussed cellular (patho-) mechanisms or assess their correctness.

Author Response

Dear Reviewer,

we are thankful for the time and effort that you spent to provide in-depth review of our review article. We corrected our manuscript according to your suggestions. Our response and corrections are listed below.

  1. I would like to commend the authors for the preparation of this extensive work.

We thank the Reviewer for appreciating our work.

  1. Systematic reviews should follow PRISMA guidelines (see http://prisma-statement.org/) While some aspects of PRISMA are fulfilled in the manuscript at hand, others are not (e.g. how were the reported studies identified,..). I recommend revisiting the PRISMA (abstract) checklist and adapt the manuscript accordingly.

            We thank the Reviewer for this advice. Indeed, we realize that the fact that our review is not a systematic review is a limitation, however, for this reason, we can not meet the PRISMA criteria, which is dedicated for systematic review.

  1. I would like to encourage the authors to expand on the discussion/conclusion and outlook regarding this topic. E.g. what should future research focus on? Where do you think lie the biggest pitfalls or potential benefits?

            We thank the Reviewer for that suggestion, we have added a sentence in the conclusion, as follows: Additional research is also needed on new therapeutic strategies, such as anti-atherosclerotic vaccines that already exist in the literature and where the biggest challenge is finding the appropriate antigen to immunize. Nevertheless, further progress in understanding and therapeutic application of the relationship between infections/chronic inflammation and ASCVD requires well-designed, large studies with long-term follow-up.

  1. Optional: shortly discussing the role of non-infectious chronic inflammatory disease in cardiovascular risk/disease might provide interesting additional insights (pathogen vs inflammation as cause for increased CV risk).

            We thank the Reviewer for this advice. We added a short discussion about chronic inflammatory diseases in the end of Section 2, as follows: Discussing the role of inflammation in the development of atherosclerosis, good models of systemic inflammation are patients with Chronic Inflammatory Rheumatic diseases (CIRD) such as rheumatoid arthritis (RA) or systemic lupus erythematosus (SLE). Patients with CIRD are at increased risk of accelerated atherosclerosis compared with a healthy population. Moreover, therapy with anti-inflammatory drugs such as anti-TNF or anti-IL-1 significantly reduces the risk, pointing to inflammation as the cause of the excessive risk of ASCVD.

Altogether, inflammatory processes involved in ASCVD are complex and require further investigation. Accumulating evidence shows that not only classical triggers of endothelial injury and inflammation but also infections may trigger and aggravate ASCVD. It remains unclear whether the mechanism by which infections affect atherosclerosis is similar to that in CIRD or whether additional factors play a role. There are also no direct comparisons between CIRD and infectious diseases in terms of ASCVD risk, thereby raising a need for additional research on this topic.

  1. Formal revision of the language of the draft might further improve the quality of the manuscript.

            We thank the Reviewer for this suggestion. We have revised our manuscript and corrected some grammatical errors and typos.

  1. Table 1 could benefit from more detailed information about the study to better inform about the quality of the studies, e.g. type of study, multicenter vs single-center, country of conduct.

            We thank the Reviewer for this advice. We added more detailed information about the studies in Table 1.

  1. I would disagree with the statement made in line 462 that Colchicine has little potential for adverse effects, especially in a vulnerable patient population often exhibiting chronic and/or acute renal failure.

            We thank the Reviewer for this statement. We added a sentence about the safety of colchicine, as follows:  However, despite its low potential for side effects, they can be a problem in long-term treatment with colchicine. In literature, the most common are diarrhea, nausea, and flatulence but there is also a risk of drug interactions with statins that increase the risk of myopathy. In clinical practice, it should be always considered whether the benefits outweigh the risk and eventually the particular population of patients that could benefit from the therapy should be identified.

We have included information mainly on gastrointestinal side effects because this is consistent with the studies cited in the article, which do not mention renal failure with this therapy. This may be due to the careful selection of patients for the colchicine study, hence we added a sentence about the need to identify patient populations who may particularly benefit from colchicine treatment.

  1. I think there is a reference missing in line 492.

            We thank the Reviewer for this insight. The entire paragraph is written based on one large study cited at the very end. However, we have added the citation in an additional place for better clarity of the paper.

Altogether, we are grateful for the in-depth revision of our manuscript and we hope that it will be considered for publication.

On behalf of all Authors,

Sincerely,

Piotr Szwed

Round 2

Reviewer 1 Report

Thank you very much for addressing my points. The decision to only consider ASCVD is well-taken so I have no remaining issue, except for these minor corrections:

  • I would suggest to switch "periodontal disease" section with "H. pylori" section to follow the physiological route (port d'entree) of GI tract. 
  • Lines 240 and 241 are exactly the same. Please delete one.
  • Similarly, lines 376 and 378 were also double.

Author Response

Dear Reviewer,
We are thankful for your very prompt reply and for appreciating our decision. We corrected our manuscript according to your suggestion. 

1. I would suggest to switch "periodontal disease" section with "H. pylori" section to follow the physiological route (port d'entree) of GI tract. 

          We thank the Reviewer for this suggestion. We modified our manuscript accordingly. 

2. Lines 240 and 241 are exactly the same. Please delete one.

     We corrected that.

3. Similarly, lines 376 and 378 were also double.
     We corrected that. 

Altogether, we are grateful for the in-depth revision of our manuscript and we hope that it will be considered for publication.

On behalf of all Authors,

Sincerely,

Piotr Szwed